# Children Are Born to Dance! Pediatric Medical Dance/Movement Therapy: The View from Integrative Pediatric Oncology

**DOI:** 10.3390/children6010014

**Published:** 2019-01-21

**Authors:** Suzi Tortora

**Affiliations:** 1Memorial Sloan Kettering Cancer Center, New York, NY 10021, USA; tortoras@mskcc.org; Tel.: +1-845-265-1085; 2The Andréa Rizzo Foundation, Charlestown, RI 02813, USA

**Keywords:** pediatric cancer, integrative oncology, integrative medicine, dance/movement therapy, dance therapy, dance/movement psychotherapy, trauma and PTSD, complementary and alternative medicine, symptom management, creative arts therapy

## Abstract

Children freely expressing themselves through spontaneous dance is a natural part of childhood. The healing powers of dance are universal in all cultures across history. Dance/movement therapy (DMT) in pediatric oncology is little known and underutilized. This article discusses DMT, specifically focusing on pediatric oncology. It defines and clarifies the difference between medical DMT as a psychotherapeutic modality aimed at addressing the patient’s psychosocial needs, and dance and therapeutic dance used recreationally to engage patients during their hospital visits. A literature review of DMT with medically ill children in the United States and worldwide is provided. It culminates with a focus on advancements in the field, discussing the future of pediatric medical DMT. Grounded in a biopsychosocial perspective, the intrinsically nonverbal and embodied nature of pediatric medical DMT is uniquely positioned to be a strong component of integrative oncology services. The use of DMT to synthesize potentially traumatic aspects of the medical experience is proposed. It ends with a call for research posing the question: Can pediatric medical DMT support the patient to express feelings while in cancer treatment within the context of a psychotherapeutic milieu, enabling the patient to create an embodied coherent narrative that fosters expressivity and empowerment?

## 1. Introduction

Children freely expressing themselves through spontaneous dance is an intrinsic part of childhood. As a child explores the world by actively engaging physically in it, a sense of self and empowerment in the world develops [1]. When children become ill with cancer, their ability to participate in playful childhood activities is frequently compromised. This loss is akin to the sense of disempowerment and loss of control of one’s body and stress physiology when facing a threatening, traumatic life event [2,3,4,5]. These experiences create a wide variety of emotions difficult to understand and express through words. Dance/movement therapy (DMT), also known in Europe as dance movement psychotherapy (DMP), can help the pediatric oncology patient regain access to this natural innate aspect of childhood by providing psychosocial therapeutic support. DMT is a body- and movement-based psychotherapeutic field that utilizes the creative expressive elements of dance to enable the patient to express feelings and gain a deeper understanding of experiences that may be difficult to speak about [6]. By creating activities that consider the pediatric patient’s emotional, social, cognitive, communicative, and motoric development, pediatric DMT provides a therapeutic holding environment in the Winnicottian sense [7,8], and reintroduces the child to her (pronouns are used interchangeably throughout the article). [Changing] body through creative dance and movement-based activities, supporting her emotional expression about her experience [9,10,11].

The generally innate pleasure in childhood of creative dance, with the sense of body control it fosters, can be especially therapeutic for pediatric oncology patients, who have the sense that their body and their disease are out of their control [11]. Nearly all treatments for traumatic stress include mastery and control over traumatic events [3,12,13,14,15]. These treatments offer solutions for the feelings of helplessness by helping the patient reimagine their body’s potential for health and comfort. This article presents the concept that offering nonverbal body and dance/movement-based solutions for the feelings of helplessness for children with medical illness, opens the potential to create a nonverbal narrative that can prevent or heal the conceivably traumatic effects of the medical experience.

The use of DMT with children with medical illness is a little known yet growing application in the field. This review article summarizes DMT with pediatric medical illness specifically focusing on pediatric oncology. The author is the senior dance/movement therapist for the Dréa’s Dream Dance Therapy Program at Memorial Sloan Kettering Cancer Center (MSKCC) within Integrative Medicine Service (IMS). The article discusses the role of DMT in pediatric oncology as an integral part of integrative medicine services. It begins with a definition of integrative oncology and shows how and why DMT fits into these services. The article clarifies the difference between DMT as a psychotherapeutic modality aimed at addressing the patient’s psychosocial needs, and dance, therapeutic dance and other recreational activities, used to engage and entertain patients during their hospital visits. The article provides a description and literature review of the state of the science of DMT with medically ill children in the United States and around the world. A world view is included to make evident the global efforts of dance/movement therapists working within this specialization. As requested by the editors of this special issue of *Children*, the article culminates with a focus on advancements in the field, going beyond history and current practice, to discuss the future of pediatric DMT and integrative oncology in the United States and around the world.

### Goals of This Article

The overarching goals of this article are to introduce pediatric medical DMT to a wider audience and demonstrate that it is an underutilized specialty, which can contribute a great deal to integrative oncology. Grounded in a biopsychosocial perspective, the intrinsically nonverbal and embodied nature of pediatric medical DMT is uniquely positioned to be a strong component of integrative oncology services.

## 2. Definitions

### 2.1. Integrative Oncology

Integrative oncology is defined as “… a patient-centered, evidence-informed field of cancer care that utilizes mind and body practices, natural products, and/or lifestyle modifications from different traditions alongside conventional cancer treatments. Integrative oncology aims to optimize health, quality of life, and clinical outcomes across the cancer care continuum, and to empower people to prevent cancer and become active participants before, during, and beyond cancer treatment“ [16] (p. 3).

### 2.2. Dance/Movement Therapy (DMT)

Body and mind integration is the cornerstone of DMT. Primarily an embodied approach, DMT enables patients to express their nonverbal felt-experiences through sensorial emotionally-rich dance and movement-based activities developed around themes of creative self-expression. These individual and group explorations create deeper conscious and unconscious self-understandings, that can transform into verbalizations, cognitive understandings and behavioral changes in daily life [17,18]. The American Dance Movement Therapy Association (ADTA) defines DMT as “the psychotherapeutic use of movement to promote emotional, social, cognitive, and physical integration of the individual, for the purpose of improving health and well-being” [19]. DMT is conducted in group, family, dyadic and individual sessions to best support the needs of the patient and the condition being treated.

### 2.3. The Orgins of “Dance” in Dance/Movement Therapy

Dance, as the first word in the name of the profession has caused some misunderstanding about the breadth and depth of the services dance/movement therapists provide. As stated by Sherry Goodill, a prominent dance/movement therapist who has spent many years developing and researching the use of DMT with medical illness, “I think there is a still an assumption about needing the capacity to move fully that keeps people from imagining DMT in the hospital settings” [20]. This belief is especially evident when introducing DMT to a pediatric patient who is not ambulatory. The word dance in the professional name has many references. Dance activities are one of many active body-awareness activities employed in the method. The field evolved from modern dancers in the forties and fifties that recognized the emotional healing powers of expressive dance [6]. Dance/movement therapists emphasize the role of dance as a form of cultural identity and acknowledge that contemporary DMT is rooted in the universal healing powers of dance in all ancient cultures across history [21].

### 2.4. Medical Dance/Movement Therapy

Goodill has built upon the ADTA definition to include, “the application of dance/movement therapy services for people with primary medical illness, their caregivers and family members…in the medical realm, DMT functions primarily as a psychosocial support intervention, complementary to conventional and standard medical treatments” [22] (p. 17). Treatment goals focus on improving experiences and addressing concerns specifically related to having a medical illness. They include: improving stress management; improving quality of life; becoming skilled in pain management techniques; reducing anxiety and depression; improving body awareness and body-self-image; decreasing fatigue; increasing vitality and energy; building a sense of resilience and hope; increasing self-care; improving social support and the ability to receive social support; and learning to accept the unpredictability of life [9,18,23,24].

### 2.5. Pediatric Medical DMT

Pediatric medical DMT is distinct from DMT used for other childhood and adolescent disorders primarily psychiatric or behavioral such as: ADHD, mental disability, anxiety, depression, and disordered eating [22]. Pediatric medical DMT serves children and adolescents suffering from chronic, acute or life-threatening medical conditions including cancer, asthma, chronic pain, migraines, scoliosis, physical accidents, Tourette’s syndrome and other neurological disorders, heart disease, and prematurity and medical complications at birth. DMT is practiced in both inpatient and outpatient hospital settings, pediatric intensive care units (PICU), and private practice clinics [11]. Adapting the definition of medical DMT to the pediatric population requires attending to additional aspects of the child’s life including: developmental considerations; a greater focus on the nonverbal communication components of DMT for assessment and treatment; and considering the family and other systems in which the child is engaged, including school and peers [9,10,11].

### 2.6. DMT in Pediatric Oncology

Cancer is a series of conditions which start with the out of control growth of cells that over-crowd normal cells, causing hundreds of site-specific diseases, rather than one specific disease [22,25]. A cancer diagnosis typically takes a profound emotional toll on the patient and the whole family. When that patient is a child, emotional reactions are often extreme. There is an overwhelming impact on all developmental stages of growth; the physical body; body image; and body mind/biopsychosocial experience of the child patient. The diagnosis and treatment demands on the family system due to the life-threatening nature and long-term high-intensity treatment protocols of a cancer diagnosis greatly affect the quality of life of the whole system. The body, mind, and emotionally-integrative approach of DMT coupled with the primary use of nonverbal and dance and movement-based expressive methods of assessment and intervention make DMT in pediatric oncology a strong psychotherapeutic method of support for the patient and the whole family system as they process this overpowering experience not easily expressed verbally [10,22,23]. The adaptability of DMT conducted in group, dyadic, or individual sessions supports both the immediate and long-term needs of the patient and family, building strong relationships throughout the full course of the treatment.

In childhood a sense of self and body image is continually growing based on the interactive experiences between self and others. Navigating medical illness during this important time of self-formation, exploration and discovery can affect the child in profound ways felt but often difficult to understand and express in words. Verbalizing these experiences can be difficult for children because they may not yet have gained the verbal and intellectual competence to speak about the complex emotions related to their illness. The emotional pain and stress of parenting a medically ill child can also make it difficult for parents to talk with their child. The role of creative expression becomes an especially salient component of DMT sessions with the pediatric population. As stated by Madden et al., “By using creative expression, a child or adolescent with cancer can express feelings about the course of the disease and tumultuous treatment through dance/movement, music, and art. This outlet allows the patient to creatively and kinesthetically process the assaults of cancer and its treatment, and thus establish a stronger sense of self and improved quality of life” [23] (p. 133).

Susan Rizzo Vincent, the founder and president of The Andréa Rizzo Foundation, a prominent organization that funds many of the pediatric oncology DMT programs in the USA uses the acronym ART to describe DMT: “I have a nice simple way of describing what dance/movement therapy is: A—The dance/movement therapist ACCESSES emotions like fear, anxiety, anger that a child with cancer may be feeling, through dance and movement expression, usually with the child taking the lead. R—The emotions are RELEASED through this movement (we all know how good it might feel to dance out our frustrations). T—a TRANSFORMATION takes place physically and emotionally through the dance and movement. Giving the child new ways to express themselves and to cope” [26].

### 2.7. Pediatric Multisensory Dance Movement Psychotherapy (MSDMT)

This author has created a specific method of DMT to support pediatric patients through medical procedures they perceive as difficult or painful, called multisensory dance/movement psychotherapy (MSDMT) [9]. MSDMT is the application of DMT with an added emphasis on the role of the body and multisensory experience to support physiologic and psychological coping, specifically related to medical illness. It supports and encourages emotional expressivity for the patient using a systematic layering of multisensory input developmentally attuned to support the pediatric patient. This includes: dance/movement, play, live and recorded music, touch, breath awareness, imagery, mindfulness, and meditation to augment pain control. MSDMT is a non-invasive treatment that compliments pharmacological and medical treatments. It is an “embodied analgesic” supporting the patient to attune to somato-sensorial sensation to engage, soothe, and empower a sense of self and coping during medical procedures [27].

MSDMT treatment focuses on engaging the whole family to create a soothing holding environment that enables both the parents and the patient to feel safe and comfortable. MSDMT is based on the premise from attachment system theory that in a secure relationship the primary caregiver acts as a protective factor whom the child [patient] goes to, to share pleasurable moments and look towards, and receive comfort from, during times of perceived danger [28]. In MSDMT the dance/movement therapist creates a sensorially-rich environment to engage the patient, family, and even the nurses and other medical staff administering the treatment. This environment can be simultaneously playful and comforting since it is created by attuning to the interests and activity level of the patient and family members moment-by-moment during the treatment. The treatment tries to help the patient achieve a relaxed, meditative state by maintaining their focus of attention on the layered sensory sensations rather than the painful sensations.

The goal of the MSDMT pain management activities is to create a multisensory environment that shifts the child’s focus away from pain, towards more pleasurable sensations. Through this process the child often drifts into a meditative or more peaceful state. This treatment includes the parents, optimizing the attachment system. Early in the development of the MSDMT activities, it became very apparent that if the parents experience their child as well cared for, they are more relaxed and available for their child, which enables the child to process his experiences with a sense of agency and control. The parents’ sense of competence is also essential. When the child looks towards his parent for comfort, he can sense if the parent is available or pre-occupied by the stress the child is experiencing. The MSDMT activities give both the parents and the child a sense of agency over medical procedures that can strip them from feeling a sense of control.

## 3. Mind-Body, Body-Mind, Bodymind, and Body-Mind-Emotion

The common thread in integrative oncology, DMT, medical DMT and pediatric DMT is the focus on the body, mind, and emotional experience from a whole person and family perspective. Focusing on practices that integrate mind and body aligns with the prevalent abolition of Cartesian mind-body dualistic thinking across medical, psychological and philosophical fields [10,22,29,30]. Eliminating this dualistic thinking causes a dilemma in discussing the relationship between the mind and the body. Starting with the word “mind” in the “mind-body” phrase can imply a directional emphasis placing the mind and cognition as the primary entrance point into the body. Mind-body techniques, such as guided imagery or mindfulness, direct the mind to notice body sensation, as a way to sense somatic experience, influence the body, and most typically to quiet the body [31]. Switching the term to body-mind emphasizes the body’s influence on the mind. In body-mind practices attention is first directed to body sensations, physical action patterns, and movements as they influence the whole body, and as an entrance point into mindful consciousness [31].

The term bodymind was first coined by Dychtwald in his book by the same name [32]. Dychtwald integrates ancient Eastern philosophy with ideas from the work of classic psychological and body-focused pioneers Wilhelm Reich, Fritz Perls, and Moshe Feldenkrais to describe the vital body mind relationship. Combining the two words as one emphasizes the body’s influence but comes closer to addressing the union of body and mind.

### 3.1. Biopsychosocial and Pediatric Palliative Care

Current research in psychosomatic illness has abandoned the mind-body and body-mind terminology, aiming to eradicate the directionality of influence between the two terms, adapting the term biopsychosocial to better represent the nonexistent differentiation between mind and body [33]. The term biopsychosocial is also used in palliative care to delineate a more holistic care approach to symptom management [34]. Pediatric palliative care as defined by the American Academy of Pediatrics (AAP) describes an integrative model providing continual care starting at the outset of diagnosis and throughout the illness regardless of whether the outcome is curative or terminal [35]. AAP pain management “…includes the control of pain and other symptoms and addresses the psychological, social and spiritual problems of children and their families” [35] (p. 351). Biopsychosocial in palliative care emphasizes there is an interdependent relationship between all three dimensions of human experience: bio, relating to the physical; psychosocial, referencing the concept of self and interactive functioning; and an implied spiritual, existential understanding related “to the meaning the patient ascribes to the illness or injury” [34] (p. 245). This definition underscores that though each dimension may to be examined separately, all symptoms occur simultaneously within all dimensions and must be reintegrated to provide holistic patient care. In their work with pediatric cancer patients, Cohen and Walco also discuss the unique DMT concept “that psychological and somatic constructs are truly identical” noting that it is only the “point of entry” that distinguishes them [10] (p. 35).

### 3.2. Body-Mind-Emotion Continuum

This author has created the term “body-mind-emotion continuum” (Figure 1) to emphasis the continuous, circular connection between each aspect of the self [36,37]. The therapeutic process begins by entering into an exploration of self through any of these aspects of self, based on the patient’s presenting focus. The dance/movement therapist may start the session with attention on the patient’s felt-experience, sensations or energy level (body). A common opening line when beginning a DMT session with a pediatric patient is, “Are you in the mood to get your body up and going or calm and relaxed?” The session can begin with a discussion of the patient’s thoughts, images and associations about their illness (mind). “How’s your day going?” Or, the patient’s emotional state may drive the intervention (emotion). “Hmm. I wonder how you are feeling today—what mood are you in today?” Each focused question is simply the starting point of the therapeutic exploration. It is not the specific point of entrance but the awareness and integration of each domain experienced through the exploration that creates true healing.

### 3.3. DMT within the Integrative Oncology Team: Cancer-WWW

Regardless of which term is used to depict the body, mind, and emotional continuum, this discussion highlights integration and includes the role of embodied experience from a whole person perspective. These methods aim to improve the quality of life for the individual, and family members and friends, creating a community of support with the health care team. Integrative oncology and pediatric medical DMT share may similar viewpoints. They both work from the perspective of integrating body and mind. They also focus on empowering both the patient and family members to be active participants in their cancer care, supporting the optimization of health and quality of life for the patient and family, across the cancer care continuum. This commitment requires team collaboration across disciplines in the medical staff.

At MSKCC, pediatric DMT is part of pediatric integrative medicine, which includes music therapy, mind-body and martial arts, yoga, and massage. The pediatric integrative medicine team works closely with child life, social work, psychiatry, nurses, doctors and the Pediatric Pain and Palliative Care Team (PACT), contributing to team meetings about patients and treatment plans bringing in psychosocial and developmental aspects of the patient’s needs. Dance/movement therapists also engage in joint sessions with other integrative medicine practitioners as well as speech and language therapists, physical therapists and occupational therapists when a psychosocial approach supports the patient’s engagement.

Both DMT and integrative medicine come from the perspective of a Whole child approach; looking at the child within the context of the experience of the Whole family system; creating a sense of community through a Whole staff collaboration—*Cancer-WWW* [27]. One goal of this article is to demonstrate why and how pediatric medical DMT is an essential member of the pediatric integrative *Cancer-WWW* team.

## 4. DMT, Child Development, and Embodied Experience

The intrinsically nonverbal and embodied approach of pediatric medical DMT embraces the body-mind-emotion continuum in a unique way that attunes, addresses and supports the [medically ill] child’s primary modes of expression and experience. Infants and young children gain experience of self and social relatedness through their active physical experience of moving and engaging in the world around them [38]. They express their feelings and develop their sense of self through their felt-embodied experiences. Stern created the term “core self “to describe the infant’s bodily experiential sense of self from birth [39] (p. 11). Core self is one of the four “sense of self” domains, from which the infant views and experiences the world from a place of reality and awareness. In this viewpoint Stern departs from the psychoanalytic view of psychic phenomena that describes this early developmental stage consisting of “delusions of merge or fusion, splitting, and defensive or paranoid fantasies” [39] (p. 11).

A more reality-based, bodily-felt experience is an essential viewpoint to take with the medically ill child of any age for it acknowledges that even young pediatric patients are present and deeply experiencing their cancer treatment. Recent research in infancy memory [40,41] substantiates the premise that even though infants and young children may not have the words to describe their experiences, these experiences can impact their development on many developmental levels later in life. Dance/movement therapists are specifically trained to attend to the patient’s nonverbal expressions as a means of assessment, expression and intervention [1,9,10,11,42]. Through sensitive observation of the patient’s nonverbal vocabulary dance/movement therapists create movement and dance activities—natural actions in the healthy child’s explorations—that provide avenues of self-expression for the medically ill child.

## 5. Dance/Movement Therapist Differentiation from Dance Teacher

It is important to emphasis the phrase “psychotherapeutic use of movement” in the ADTA definition to clarify a frequent misunderstanding for those not familiar with or only observing DMT in action. Due to the physical nature of DMT, it is often confused with other movement and body-focused activities used to promote wellness and wellbeing such as yoga, mindfulness, martial arts, contemporary dance, improvisation, ballroom dance, Broadway theater, jazz dance and therapeutic dance. These programs are fun, creative and often provide physical skill training taught by a professional teacher in the specific field. Such activities include learning specific dance choreography, yoga postures, martial art moves, or meditation breathing techniques to focus one’s attention. These are extremely useful skills supporting the cancer patient to feel a sense of competence and control of their body. Many dance/movement therapists are also trained in these techniques and include these activities as part of their therapeutic repertoire.

The core difference between the activities described above and DMT is that DMT is a form of psychotherapy with an added experiential physical creative arts focus. Using the body enables rich nonverbal psychic material to emerge supporting embodied discoveries along the body-mind-emotion continuum [36,37,43]. The creative arts dance and movement aspect of the work occurs within the psychotherapeutic container created by the therapist witnessing and moving with the patient. It is the psychotherapeutically supported personally aesthetic and deeply personal creative explorations, which uniquely distinguishes DMT from other mindfulness, body and dance forms. In DMT the patient uses physical activities to convey emotions creating psychological connections between their nonverbal and verbal expressions. Through DMT the patient explores how their body, mind and emotions work together to express their feelings, while gaining coping skills facilitated by the embodied explorations.

## 6. Dance/Movement Therapy Educational Training

Many dancers pursue further training to become dance/movement therapists due to their own experiences of the communicative emotionally empowering aspects of dance as dance performers and/or dance teachers. In the USA there are two tracks to becoming a dance/movement therapist. Dance/movement therapists can get a master’s degree from a university program accredited by the ADTA. If they have a masters in an allied field such as mental health counseling, psychology, marriage and family counseling, social work, or dance they can pursue the alternate route track through the ADTA by taking specifically-required supplemental courses in DMT and psychology taught by ADTA approved dance/movement therapists and other professionals. Many dance/movement therapists have additional licensing in psychological fields including mental health counselors, marriage and family counselors or social workers. A growing number of dance/movement therapists have earned a doctorate in such fields as psychology, education, infant mental health or dance/movement therapy.

## 7. State of the Art of Pediatric Medical DMT—The Specifics

At MSKCC, DMT happens everywhere—in the inpatient room, outpatient play room, and even in the hallway. Sessions occur individually, with parents, family members, friends, and groups of patients. Even the nurses and doctors join. The goal is for the patient and whole family to feel that the dance/movement therapists are available and present anywhere at any time.

This author designed the MSKCC program to be all encompassing. It provides an avenue for the child to express his or her needs, concerns and experience with cancer treatment through the supportive creative and aesthetic medium of self-expressive dance, music, play, relaxation, and talking, by meeting the child wherever he is in the moment, along the emotional cancer continuum. Each session follows the child’s lead. One day the child may want to gleefully dance exploring the joys of moving, while another day the same child may feel listless, needing to share a sense of quiet contemplation with the dance/movement therapist. These child-directed dance-play sessions build the pediatric patient’s sense of competence, mastery, and autonomy on a very embodied level.

A core premise of the program comes from this author’s understanding that all infants enter the world and accumulate experiences first and foremost through bodily sensations, reactions, expressions, and experiences, developing a “sense of body” that informs the development of a sense of self [1] (p. 31). Since these body experiences, define and continually inform the child about who he or she is, it is essential that the pediatric patient’s felt-embodied experience is taken into account. It is critical that the dance/movement therapist tries to detect who the patient “is” beneath this layer of illness, determining how the medical experience may be coloring the patient’s pre-existing sense of body, and subsequently influencing the patient’s sense of self. The dance/movement therapist strives to help the patient experience themselves separate from the confines of the illness. Dance/movement therapists training in nonverbal analysis greatly informs how they analyze the movement actions to gain a deeper understanding of the patient’s self-expression and felt-experience.

### 7.1. Pediatric Medical DMT and Nonverbal Analysis

As stated by Mendelsohn, “Dance/movement therapists believe that the body “speaks” and it is our task to decipher what the body behavior of hospitalized children is trying to communicate in order to help medically ill children cope better with their illnesses and hospitalization” [42] (p. 65). Deciphering nonverbal bodily expression is at the core of DMT practice used in both assessment and intervention [1,10,44,45,46,47].

Dance/movement therapists train in one or more nonverbal analysis systems, with Laban Movement Analysis (LMA) [44] and Kestenberg Movement Profile (KMP) being the most prominent. LMA is the original nonverbal analysis method, developed by the modern dancer, Rudolf Laban, in the 1960s [48]. It provides a way to analyze the qualitative and quantitative components of action and the processes involved in executing movements rather than simply describing the goal of the movement [44]. The KMP, developed by psychiatrist Judith Kestenberg, also in the 1960s, is “a Laban-derived method” to categorize actions that includes a development and psychological interpretation of the mover’s movement repertoire [49] (p.5). Through these systems dance/movement therapists observe the qualitative aspects of the patient’s movement patterns to gain understanding of how their body actions may reflect their coping style and emotional state [50]. Through nonverbal analysis the dance/movement therapist assesses the level of physical engagement the patient can sustain, the theme and symbolic nature of the creative expressive dance activities and the patient’s intra- and interpersonal functioning [9,10,47].

### 7.2. Laban Nonverbal Movement Analysis: Short Description

This author is trained and certified in both LMA and KMP. A thorough description of each system and their categories goes beyond the scope of this article, so a simple explanation of the LMA system is explained to provide clarification for the case vignettes that follow.

The qualitative elements of Laban nonverbal movement analysis are described through five movement categories: Body, Shape, Effort, Space, and Phrasing [44,48,50]. The Body category includes what areas of the body are used most frequently to determine the sense of body coordination and sequential movement patterns. The Shape category reveals how the position and shape of the child’s body reflects their experience in the moment of their own body, and how they position themselves in relationship to others. The feeling tone of the actions includes the use of weight, spatial directional focus, timing, and the tension and flow patterns of the movement, described as Efforts. Space reflects how one’s body takes up space, and moves through space, including spatial levels and spatial pathways. Phrasing describes the rhythmic, arrhythmic, syncopated or melodic phrasing of the mover’s actions.

### 7.3. Socially-, Religiously-, Culturally-, and Ethnically-Sensitive Practice

Attuning to nonverbal expression also enables the DMT to be sensitive to social, religious, cultural, and ethnic differences communicated through actions, gestures and rhythmic patterns of the individual patient and family member, as well as the group dynamics expressed nonverbally. This is especially important at MSKCC where the specialization in specific and rare childhood cancers such as retinoblastoma and neuroblastoma, bring families from all around the world for treatment. Observing, attuning, and responding to the qualitative differences expressed through nonverbal postures, actions and gestures enables the dance/movement therapist to learn about the pediatric patient while being sensitive to the family dynamics within their culture. The dance/movement therapist uses this information to create an environment that is supportive and familiar to the patient and family. Understanding social, religious, cultural, and ethnic difference is especially important to support the patient and family during palliative and end of life care.

### 7.4. Embodied Countertransference

Though the DMT strives to attune to and be sensitive to the unique characteristics and behaviors of the patient and his family, the therapist is viewing and interpreting these behaviors from her own lens of personal experiences. The term countertransference is used by psychoanalysts and psychodynamic therapists to describe thoughts, feelings, reactions and interpretations the therapist has that may be triggered by her own life. Dance/movement therapists consider their bodies and movement actions as essential tools in the treatment. Attention to their personal bodily-felt reactions, sensations and responses is a core process in the embodied countertransference experience. A thorough description of how dance/movement therapists explore countertransference goes beyond the scope of this article and can be found elsewhere [1,51,52]. Simply stated here, dance/movement therapists pay attention to their thoughts, images and associations within the context of their felt-sense reactions on a body and emotional level. Dance/movement therapists actively use their own bodies to sense the patient’s nonverbal experience while observing and moving with the patient. Understanding the personal lens that may influence the therapist’s responses plays an important role in how the session unfolds. The dance/movement therapists Plevin and Parteli describe their use of nonverbal analysis on the onco-hematology unit to determine how they structured their interventions and how they attended to their own embodied countertransference responses during the treatment [47].

## 8. What Does DMT Look Like?

To exemplify the elements of pediatric medical DMT, this section provides case study vignettes organized by age group, demonstrating the changing developmental considerations. The first vignette is the longest example that provides a full overview of the DMT experience. It presents a thorough description of the nonverbal LMA approach within the context of the dance/movement therapist’s embodied countertransference and cultural considerations when working with a parent-infant dyad. The remaining shorter vignettes offer additional examples to portray other aspects of the work. Each vignette is a composite. All names and identifying specifics of patients, including details about family members or other social details, have been changed to protect their privacy.

### 8.1. Parent-Infancy Dyadic DMT—Four Session Description

As I enter the inpatient room of Jae-joon, a nine-month-old with choroid plexus carcinoma (this is a rare malignant brain tumor that occurs most often in children under the age of 2 years old) I am instantly struck by the emotional stillness I experience in the room. Jae-joon is sitting in a forward concave posture in his crib, staring motionless at his mother, who is sitting lower down on the other side of the crib, on the edge of the sleeper couch. She is also in a concave posture with her elbows on her knees, holding her head in her hands. As I come closer to the crib to introduce myself, Jae-joon immediately tenses his whole body, quickly folds his torso over his legs as he vigilantly attempts to reach his arms through the crib bars towards his mother, seeming to attempt to get away from me. He audibly whimpers with a strained cry. Mom does not move. In my own embodied countertransference, I sense my breath tighten as my stomach clenches and a feeling of deep sadness and isolation comes over me. “Does Mom not hear her baby’s cry?” I wonder. I take a few long breaths to soften my own bodily stance and pause too. I review what I know in my mind. Mom conceived this baby at age forty, after waiting to conceive for five years, due to her previous birth of a baby who died of the same condition.

The emotional anguish in the room is palpable. Jae-joon, at age nine months is rightfully at the height of stranger-anxiety, which can only be amplified by his mother’s obvious agony over the potential loss of him. During this first visit I speak softly to Jae-joon without moving closer, to respect his attempt to spatially distance himself from me, assuring him I am only here for play and comfort. I will perform no medical procedures on him. I slowly walk over and quietly stand next to Mom keeping my arms close to my torso without gesturing on respect her private contained spatial stance. I gently explain my role as the dance/movement therapist. She does not look up, but softly moans and rocks her body forward and backward almost imperceivably. I crouch down, next to her while still maintain spatial distance, and am at eye level to Jae-joon without staring at him. I slowly rock my body and hum to the rhythm of her actions. Mom looks up at me and for a moment our eyes meet, tenderly. I delicately state I understand how hard this is and tell her I am here to help.

When I return for the next session Mom and Jae-joon are again in the same positions. Though, this time Mom looks up on her own when I come to her side. Jae-joon pays attention to this, giving me a quick glance when he sees his mother look towards me. I ask Mom how she and the patient are doing. Mom looks at me stating something in a heavy accent I cannot understand. I smile and show her my music selections attempting to indicate I want her to show me some Korean music they are familiar with. She does not understand, so I put on a classic Korean lullaby and softly sway to the rhythms. Jae-joon maintains his vigilant stance but watches as mom looks at me and smiles. She does not seem to notice Jae-joon looking at her. I am patient with this, for I know if I can make Mom feel more comfortable, she can be more present for her baby. She faintly nods to the musical beat and I mirror her actions. Jae-joon attentively watches this exchange with less vigilance, though he does not move. As our session ends I feel lighter and Mom, though still quiet, also appears more relaxed. Our shared dance has had a calming effect.

Though the medical records state that Mom understands English, I bring Juhee, a Korean colleague with me to the next session. I am wondering if the baby might feel more comfortable seeing me with Juhee and if Mom will open up in her native language. I also ask Juhee to give me the names of current popular Korean children’s songs, which I download on my iPhone before the next session. When I enter Juhee takes the lead speaking in Korean to introduce herself. This time Jae-joon does not immediately avert his eyes towards his mother but looks directly up at Juhee. Mom stands up and softly welcomes Juhee. Juhee explains in Korean what DMT is. She asks Mom about music they like to listen to and fortunately I have that song downloaded. As I put the familiar song on, both Jae-joon and Mom look up and naturally sway to their familiar beat. In this way, we all dance, in our private spaces and the solemnness of the room dissolves. The session ends with Mom sharing more details about her feelings and her concerns and sadness about Jae-joon’s illness. She specifically states that Jae-joon had been crawling and pulling up to standing but has dramatically regressed, not having the desire or capability to move.

When I return alone for the next session Jae-joon does not lean over to his Mom but greets me with an expression of curiosity. Mom stands up and comes over to me with a friendly sway to her step. I put on the same song and this time encourage Mom to hold Jae-joon as we dance together. I want Jae-joon to experience the warm body-to-body contact of his mom moving in a rhythmic manner to support their emotional relationship and to stimulate his own physical awareness. When the recorded music stops, Mom sings her own favorite song to Jae-joon, and Jae-joon softens even more in her arms as he looks up at her face with a smile. From this session on, we begin each session with Mom’s singing to Jae-joon, who becomes more and more engaged both emotionally and physically. He pulls himself up in his crib reaching his arms out to be held. His physical, emotional and social skills improve simultaneously as Mom is more emotionally available to him. During these dance therapy sessions Mom also shares more of her feelings and becomes more responsive to receiving support from the whole staff.

### 8.2. Preschool-Age Individual DMT Session

James, age four is sitting in the outpatient playroom with his Mom when I see him. It is eleven o’clock in the morning and he has not had breakfast due to his upcoming procedure that requires general anesthesia. I know him well for James has been coming to MSKCC for retinoblastoma treatment (retinoblastoma is a rare childhood cancer that most often effects infants and young children, occurring in the retina of the eye. It may occur in one or both eyes), for over a year and the medical treatments have been difficult for him to handle. I am drawn to him today for I notice his restless body actions and mournful exclamations, despite Mom’s efforts to read a book to him while awaiting his procedure. As I walk over Mom, looks up, with a pleading expression, indicating this has not been an easy morning. I ask James if he wants to dance and he jumps up readily. When I ask what music he wants to dance to he recites the words to the classic children’s song, “We’re going on a bear hunt” [53]. I put on an instrumental piece of music with a syncopated drumbeat and melodic strings creating a sense of suspense to support the adventurous tone of the storyline. I join James in reciting the first lines of the story, complete with the gestures miming the action words. “We’re going on a bear hunt. We’re going to catch a big one. What a beautiful day!” I mirror James widespread arms to depict catching “a big one” and then reach way up high to feel the beautiful day. This is followed by placing our hands on our hips and shaking our heads “No,” to demonstrate our lack of fear, in response to “We’re not scared!” But suddenly the danger sets in. “Uh! Oh! Grass, long wavy grass.” Quickly our eyes open wide and our hands go to our mouth in mock surprise. “We can’t go over it. We can’t go under it. Oh no! We’ve got to go through it.” We arc our arms up and over, then down and up, followed by a determined stretch forward as far as we can reach. We perform these gestures repeatedly, as this refrain repeats between each new obstacle that comes our way. We must pass through a deep cold river; step with conviction into thick oozy mud; stumble and trip within a big dark forest; and spin and sway through a swirling whirling snowstorm. Making ourselves small, we tip toe into the cave. We come face to face with the bear, which briskly sends us pantomiming back through each obstacle until we are safely under our covers in bed.

As we encounter each new hurdle, James’ actions become more determined and stronger, taking up more space, as he recites the words of the refrain with increasing conviction. As I mirror his actions, matching his mood and tone, I am taken by the metaphor of the words. The message is clear…”We’re [trying to] not [be] scared! … But… Uh! Oh! There is no choice.…We can’t avoid it [the cancer experience]. We must go through it [and find a way to bravely approach it].” My heart beats with a quickened pace and I look up to see the dad of a teen I have also worked with, watching us. As our eyes meet, I see tears glistening in his eyes. I smile warmly, privately sensing my own emotions quietly welling up too. I know, he knows the journey we are dancing out through our embodied drama. James soldiers on with renewed strength evidenced in the clarity of his body actions and eagerness to repeat this song repeatedly. The details of each stanza pale in comparison to his enactment of the refrain, which become bolder with each repetition. His bravery surges as the bear hunt becomes a lion hunt. In my attuned felt-experience I understand the stakes are getting higher. I ask him if he wants to hear a lion roar and he eagerly agrees. I happen to have some recordings of a lion roaring. I put it on. James shudders, for just a moment and whispers, “That’s scary!” “Yes” I say with compassion, “but what are we going to do?” James does not miss a beat and shouts as his actions get even stronger, “We’re not scared! We can’t go over it. We can’t go under it. Oh no! We’ve got to go through it.” Our dance continues. When I tell him this is our last round, he adds a tiger to our hunt, singing out even louder, with his gestures passionately matching his words. James is beaming, standing tall and proud when we finish our dancing pantomime. As I reflect upon my experience with James I am touched by his wise soul. He knows without knowing, just the right song he needs to add courage to his upcoming treatment. Beaming, Mom thanks me for the transformation that has taken place, as James snuggles cozily with her now.

I walk over to the dad watching. He takes my hand, thanking me for today, for the opportunity to experience the tender beauty of the moment. He thanks me for the courage and joy that dance/movement therapy awakens. He shares that watching our dance evoked his experience with his own son at that age, when he was getting his cancer treatment, from which he is now in remission. He knows the journey ahead for this boy and his family. At that moment his own teen son comes over with a bounce in his step and his loving gaze of gratitude and pride is palpable.

### 8.3. Music and Dance/Movement Therapy Group Session—Elementary Age

Every Monday morning Karen Popkin and I offer a music and dance therapy group in the pediatric ambulatory care center (PACC). The group is attended by children of all ages, their siblings and other family members. For many, it seems to ease the transition back to the hospital after the weekend break at home. Music and dance therapy, both part of the creative arts therapy field (along with art, drama and poetry therapy) are wonderfully complementary. Through Karen’s intuitive live guitar improvisations and my attuned embodied actions in response to Karen and the participants spontaneous music making we can draw out each participant to access their emotions through music, dance and their imagination.

On this day we have seven participants. Three patients ranging in age from four to seven, two siblings approximately age five and eight and two patients’ parents. At this point in the session each child is playing an instrument, choosing from Karen’s assortment of tambourines, egg shakers, bells, rainsticks, and drums. One child starts an even paced beat and Karen strums along holding the rhythm steady. One by one each participant adds to this beat creating a rhythmic symphony. I notice Pierre, age seven stamping to the beat as he sways his body forward and backward in his chair. I mirror his actions with precision and ask him, “Let’s go on a journey. Where are we going?” “We are in the ocean, there is a storm” he shouts! Immediately the pace picks up as the other participants excitedly join in. Clara, age five, jumps up, “I fell off the boat” she exclaims as she feigns falling. Two more children stand up, swaying to and fro, arms flailing, as if caught in the waves, while the remaining children add more vigor to the musical accompaniment. “A shark” someone calls out and instantaneously all the children begin to scutter around as sharks lunging zealously toward imaginary bait. I pull out a large blue sheer scarf and the parents help me hover it over the children, who dart and dive and dip, exploring this underwater scene, spurred on by the eager musicians, accelerating the beat. At some point the waves part as the rainstick and melodic guitar take over the percussive beat. The shark children slow their bodies as they begin to stretch out, “floating” above the surface of the water and then settle back on their boat-chairs. We discuss how powerful the sharks are, how brave they are, how they endure the storm to capture their prey and how good it feels to be safely back on shore. The concrete and the symbolic merge here as the children explore both the scariness and the potency the shark imagery evokes in them. The shark metaphor is a common image parents use with their children, to depict how the cancer treatment is attacking the bad cancer cells. Again, I marvel at how through this imagery created by the children they confront their fears, embodying both the aggressor and the warrior and then, savor in their success.

### 8.4. Adolescent DMT Individual Sessions—Summary

Dimitri, age 16, recently received a Hodgkin lymphoma diagnosis (this is a cancer that effects the lymphatic system and starts in the white blood cells). As a high school basketball player, he is referred by Child Life to DMT to provide support for his active lifestyle. He is surprised but welcoming, “Hey I’ll try anything at least once!” he says with a warm smile. He visibly relaxes even more after I explain that DMT is more than dancing. We settle on having him show me some of his basketball warm-up activities. Once out of bed I look up at his tall lean yet muscular 6′2′ frame and take a breath. Through my own kinesthetic empathic countertransference, I experience his current physical strength and ponder about his potential vulnerability. I imagine the journey ahead and wonder what imagery will work best to keep him moving, even if on some days it is only figurative. I ask about music choices and to my surprise he is a “High School Musical” fan. We agree on “Get your head in the game!”—an apt choice for a basketball player. As Dimitri reaches his arms way up high arcing over to his left and then right, the high energy, upbeat rhythms and focused message of the song directs our pulsing actions.

Dimitri continues to take the lead, demonstrating his agility as he pumps his legs up one at a time and slaps his knees. As he becomes more limber, he also opens up about his recent experience with his illness. Between stretches, in which he seems to be unconsciously testing what his body could do, and what it can do now, he tells me he is the captain of his team. One of his jobs as captain is to lead the warm-ups. It was during such a warm-up that he suddenly felt weak and collapsed. He actually hadn’t been feeling like himself for a while, but thought it was just the lack of sleep due to his heavy basketball schedule and trying to keep up with his homework. In my active following, I experience Dimitri leading his whole team, keeping them challenged to keep them going. I playfully ask him about this and he laughs in agreement. He discusses how much he enjoys physical challenges. He pauses and looks directly at me. We each take a deeper breath, spontaneously. This opens up a discussion about his cancer treatment plan. He wants to know what to expect; he wants to be included in decisions; and he wants to do everything he can to stay active. We plan for regular sessions and I explain there are many ways to keep him active, even when his body may not be as energetic as he is used to.

During his treatment, the team keeps Dimitri informed, and he stays true to his word with me. Even on days of active chemotherapy when his red blood cell count is low, he bruises easily, and he is feeling extreme nausea and fatigue, he welcomes me with a faint smile. Over the months of our work together we create a full playlist of songs to fit his different moods. Some uplifting and hopeful, some wordless tunes that induce quiet contemplation. Some days he is chatty, some days we can do a version of his stretches, other days, his fatigue takes over and he participates through guided visualizations we created during our sessions over time. Throughout our work, Dimitri rises to the challenges his cancer treatment present, as he learns how to attune and dialogue with his mind, body and emotional messages.

### 8.5. Pediatric Medical DMT and the Caregiver-Infant/Child Relationship

As stated in Section 3.3, an intricate part of the Dréa’s Dream Dance Therapy Program as practiced at MSKCC is to support the whole family as they navigate their experience of cancer treatment. Providing support for family members and other significant caregivers takes many forms depending on the age of the patient, the needs of the caregivers and the patient’s stage of treatment. At the most primary level, caregivers of patients of all ages experience great pleasure and relief in seeing their child participate in an active, joyful and creative physical activity that is such a typical part of childhood. This enjoyment turns to gratitude quickly as most caregivers see the underlying emotional expression that the DMT activities foster. They frequently request the presence of a dance/movement therapist to support the patient before, during or after a difficult procedure.

Though the child treated for cancer is the designated patient the dance/movement therapist pays keen attention to the needs of the caregivers focusing on the significant role they play in the patient’s overall wellbeing. Psychosocial supports for the caregiver include verbal therapy specifically addressing the caregiver’s immediate concerns, sorrows, experiences, and personal associations that arise from their child’s illness. Dance/movement and mindfulness activities are created to also support the caregiver’s nonverbal expression. Psychoeducational information focusing on infant, child and adolescent development is an integral part of the treatment as well. These sessions can occur separate from the patient or within the context of a patient session.

Caring for the caregiver in the context of the patient’s treatment also strengthens the child-caregiver relationship [54]. With infants and toddlers, the caregiver is encouraged to participate in activities fostering the patient’s felt-experience of their parent, literally and symbolically providing a secure emotional base. Caregivers are often hesitation and fearful to physically engage in playful ways with their medically ill babies. Creating safe activities together that foster a natural body-to-body connection filled with loving touch rather than anxious tense touch re-connects baby and parent on a deeply emotional level. Building activities that support interactive engagement strengthens the primary attachment system. In this growing era of even the youngest patients showing intense interest in electronic devises and visual media, preserving the emotional embodied live parent-child relationship as the primary source for pleasure and comfort is essential.

Minde and other infant mental health researchers provide evidence that interventions directed at supporting the parent–infant relationship during an infant’s early hospitalizations build the parent- infant interactional patterns and psychological functioning of the infant [14]. Minde highlights the importance of interventions that allow the infant to direct and modulate the intensity of the free play with the caregiver. This promotes a sense of interactive agency, security, and competence that the child does not always have within the context of his medical treatment relationships.

As the pediatric patient reaches pre-school and elementary age, caregivers become both active participants and audience members, depending on the child’s requests. During this stage of growing independence, the child’s sense of agency blossoms as they direct their own dance-play or choose to include their caregivers or even the nursing or medical team. Themes of these storylines enable the patient to enact a sense of control over their bodies and their medical experience, in marked contrast to the adult-driven medical procedures they actually go through. The medical team’s willingness to participate becomes another way to strengthen their rapport with the patient. This playfulness and sense of trust carries over during subsequence medical procedures. At this age, parents also take pleasure in seeing their child build multiple secure relationships.

During adolescence, the dance/movement therapist has both a separate and joint relationship with the patients and their caregivers. At times the whole family engages in activities and verbal conversations. Often caregivers at some point leave the room to give the patient private time with the dance/movement therapist. Interests and concerns, including ways to support the patient’s perspective that may be different than the family’s views are addressed.

Supporting the whole family system is especially important during end of life care. Activities are created to both sooth the patient’s experience and help the caregiver’s cope with this next stage. Rituals are created, building upon the rapport developed between the dance/movement therapist, the patient and family members through their work together. Memories of favorite songs and activities are incorporated into these sessions, providing family members with the opportunity to express their emotions, while holding the patient’s needs as well, to support a peaceful transition.

With all age groups the overarching goal of the DMT treatment is to provide a safe place for each patient’s expression of their embodied experiences within the context of supporting the whole family. These experiences may be felt and known or felt and unknown on an emotionally conscious level. Providing support with awareness of the dynamics of the whole family can greatly augment the whole cancer treatment.

### 8.6. Pediatric Medical DMT within the Context of the Medical Team

The dance/movement therapists at MSKCC work closely with many members of the medical team, participating in meetings to plan treatment goals and pre and post-operative care, writing chart notes and working side-by-side during treatments. A dance/movement therapist attends weekly inpatient pain and palliative care (PACT) meetings participating in discussions about how to support the patients to cope with their medical experience; increase engagement and activity level; or transition to end of life care. The nonverbal aspects of support DMT provides is especially integral in helping patients who are too young, or reticent to talk about their experience; exhibit resistance or anxiety during treatments; or are overly reactive and sensitive to routine procedures such as dressing changes, taking medicine, getting shots or finger pricks. In these situations, the dance/movement therapist typically meets with the medical team to learn details about the case. She presents her ideas about the psychological themes that might be underlying the behaviors, which can include information gleaned from previous sessions with the patient. She has a session with the patient, writes-up the session in the patient’s medical files and follows up with the medical team to share any important shifts that occurred during the treatment.

The dance/movement therapists work very closely with the child life team providing group dance therapy sessions that support the many year-round themes and events the child life staff organize such as sibling day, the prom dance and many holiday celebrations. Each member of the child life team is also assigned to work directly with patients with specific diagnoses. The dance/movement therapists often organize treatment goals and work in tandem with specific child life specialists to create continuity in the patient’s psychosocial care. This author has used her expertise in infant mental health to help child life staff create charts for the patient’s room outlining age appropriate developmental activities to support the infant or toddler’s social/emotional engagement.

The MSDMT activities discussed in Section 2.7 were created specifically to support the family and nursing staff during the administration of an antibody treatment for high-risk neuroblastoma [9]. MSDMT is used as a nonpharmacological support alongside of the pharmacological supports, to help reduce the patient’s anxiety and side effects of the treatment, which include pain [55]. The MSDMT activities evolved the first year this author was working at MSKCC when she was asked by a nurse administering the antibody treatment, to assist her with a toddler that was exhibiting extreme stress. Upon entering the room, the dance/movement therapist surmised that if she could create a soothing environment to easy the toddler, and by natural association the parent’s distress, the nurse would be able to administer the treatment more efficiently. The success of this partnership between the nurse and the dance/movement therapist is evidenced in its continuation over fifteen years later. Nurses and parents request the support of the dance/movement therapy team during the administration of the antibody treatment, each dance/movement therapy intern learns the activities and other integrative medicine and child life staff have incorporated the activities into their support activities as well.

## 9. Prevalence of DMT in Pediatric Oncology and Other Medical Illnesses

To prepare for this article, the author reached out through phone calls and emails to the ADTA and European Association Dance Movement Therapy (EADMT) email listserves; colleagues and leaders of DMT organizations; and DMT researchers to gather information about past and present pediatric oncology and other medical DMT programs. She also conducted a literature search about DMT in pediatric medicine from 2007–2018. Creative arts therapy programs not including a dance/movement therapist on staff were excluded from the search, to delineate DMT programs and dance/movement therapists employed in pediatric medical DMT. Since this was not an exhaustive or systematic search based on a research protocol, there may be unintentional omissions. It is complete only within the author’s knowledge and responses received.

Despite the natural fit of DMT with the pediatric population, the underutilization of DMT in medical settings is profound [56]. Of the 120 citations in PubMed, Embase, CINAHL, and PsycINFO with search strategies using variations on dance therapy with infants, children, and adolescents only three articles are specifically about medical DMT. Of those only two are specifically about pediatric oncology [9,23,56]. This author’s own library of medical DMT literature spanning 1993–2018 adds seven literature sources, with only two specifically about DMT in pediatric oncology [10,11,18,22,34,42,57].

Research in pediatric DMT oncology is even more sparse, though this is not surprising because research in all DMT populations is lacking due to several factors. The obstacles for DMT research include: the lack of manualized treatment methods due to the creative improvisational and patient-driven nature of the clinical work; the lack of consistent, valid standardized outcome measures; verbal intervention tools do not accurately measure the nonverbal body-oriented interventions; many of the studies are pilot studies with small sample sizes without larger follow-up studies; and there are very few research faculty positions for DMT worldwide [11,18,24]

The dance/movement therapists, Sherry Goodill, Robin Cruz, and Jennifer Frank Tantia in the USA; Sabine Koch, Marianne Eberhard-Kaechele, Helen Payne, Iris Braeuninger, Vicky Karkou, Bonnie Meekums, and Rosemarie Samaritter in Europe; and Elizabeth Loughlin and Kim Dunphy in Australia have helped to address this research dilemma by developing new research strategies to encourage colleagues and masters and doctorial DMT students to undertake research. This list of dance/movement therapy researchers is based on this author’s current knowledge and may not be complete. Any omissions are accidental.

In a recent investigation for specific research about DMT using the search terms cancer, psycho-oncology, and oncology from 1960–2017, Goodill found ten studies, with only one in pediatrics—pediatric brain tumor patients receiving outpatient chemotherapy [18,23]. A common factor in all the studies Goodill found, which include participants both actively and finished with treatment, is a low dropout rate of participation, suggesting that patients experienced benefits that included improvements supporting quality of life. A meta-analysis by Koch et al. found similar results [24]. The study states that participants experienced a decrease in depressive symptoms and anxiety leading to stress reduction. The participants had a greater sense of vitality and a sense of self-efficacy, predicting higher-levels of adaption and symptom management. The creative expressive process, which included affirmative images, metaphors, and symbols enabled the participants to find inner resources and experience personal strengths.

### 9.1. Pediatric Medical DMT in the USA

One of the earliest articles published in 1993 about pediatric medical DMT is written by Goodill and Morningstar [11]. The article outlines the principles of DMT including the use of movement as our primary communication, the importance of the creative process and symbolic expression and the dynamic aspects of body image. A list of the psychosocial goals of DMT, include: decreasing anxiety related to the hospital experience and medical procedure; help the pediatric patient adjust to functional changes in the body and body image; support the patient’s active experience of their body; and supporting the child’s expression of their illness and hospital experience, rather than focusing on their dysfunctions caused by the disease. Goodill focused her doctorate on medical DMT with all illnesses of all age groups. This became the basis of her 2005 book [22]. She is now a primary researcher in the field striving to build the efficacy of DMT [18,58]. Her 1993 article and her book are key references used in all subsequent publications about medical DMT.

The recent growth of DMT oncology in pediatrics is largely due to the generous support of The Andréa Rizzo Foundation [59]. The Andréa Rizzo Foundation is a 501c3 non-profit that funds Dréa’s Dream Dance Therapy Program for children with cancer and special needs in hospitals and schools nationwide [60]. Susan Rizzo Vincent, the founder and president of the foundation, began The Andréa Rizzo Foundation after the tragic loss of her daughter, Andréa, at age 24, who was struck by a drunk driver [61]. Andréa, a cancer survivor herself, cured from neuroblastoma at MSKCC as a baby, had the dream of starting a pediatric DMT program in MSKCC. She was taking preliminary DMT courses at the time of her death in 2002. Through the tireless support of her mother and friends, The Andréa Rizzo Foundation was founded, with MSKCC as the inaugural Dréa’s Dream Dance Therapy Program in 2003. Starting with three hours a week, it quickly grew to a thirty hour a week schedule including two part-time dance/movement therapists and an internship program. In addition to the DMT school programs, The Andréa Rizzo Foundation now supports pediatric hospital programs in eight hospitals ranging from monthly visits to our thirty-hour a week program. At the time of this writing the hospital programs include: Children’s Hospital Los Angeles; Connecticut Children’s Medical Center; SSM Cardinal Glennon Children’s Medical Center in Missouri; Levine Children’s Hospital in North Carolina; Hasbro Children’s Hospital in Providence, RI; MSKCC in New York; Ronald McDonald House of New York and St Mary’s Hospital for Children also in New York.

The following hospitals also have pediatric medical DMT programs for acute and chronic medical conditions. These are added based on responses to the email inquiry: a part-time DMT internship program at RWJBarnabas Health, in New Jersey; a full time DMT program in the Child Life and Expressive Therapies Department of Children’s Hospital of Wisconsin [62]; a part-time dance/movement therapists and internship program at Children’s Hospital of Philadelphia (CHOP); and a DMT internship program at St. Christopher’s Hospital in Philadelphia.

Once a thriving medical DMT program for pediatric oncology-hematology, the child life/creative arts therapy program at the Joseph M. Sanzari Children’s Hospital, at Children’s Cancer Institute, Hackensack University Medical Center in New Jersey (formerly The Tomorrows Children’s Institute) is omitted on this list for [to the best of this author’s knowledge] it has no dance/movement therapist on staff. This program which began 1987, was first supervised by an art therapist. In 1995, Orkand (formally Cohen), became the first dance/movement therapist to become the supervisor. She also became a certified child life specialist. Orkand supervised this program, which included and extensive DMT internship program from 1995–2007. In 2007 she was hired by the pediatric pain and palliative medicine department to bring her DMT skills in attending to the nuances of communication to this aspect of patient care. In this position, she became “…a weaver, taking a wholistic approach to listening to patients and their family’s stories attuning to help the medical team attune to their style of communication and decision making” [63]. Orkand’s writings (written with the surname, Cohen) about the role of medical DMT in pediatric oncology-hematology and pain and palliative care are also widely read [10,34].

### 9.2. Inaugural Pediatric Creative Arts Therapy Conference at CHOP

This past April 20–21, 2018, The Creative Art Therapy Department at CHOP organized its inaugural Pediatric Creative Arts Therapy Conference. A call for proposals was distributed nationwide. Of the eighteen presentations three were about pediatric medical DMT. Of those three, one was presented by Erin Gallagher, the dance/movement therapist at CHOP, discussing the benefits of DMT with a range of acute and chronically ill pediatric illnesses. The second was by Jocelyn Shaw, a DMT and psychologist working at The Ronald McDonald House of New York, funded by The Andréa Rizzo Foundation. With Jessica Horn, a research nurse at CHOP, they discussed collaborative care between DMT and nursing. The third presentation, also representing a program funded by The Andréa Rizzo Foundation was by this author and DMT colleague Jennifer Whitley at MSKCC, discussing the core elements of our MSKCC pediatric medical DMT program, to best support cancer patients and their families.

### 9.3. Pediatric Medical DMT Worldwide

At this time to the best of this author’s knowledge, based on email inquiries, there are no current pediatric medical DMT programs in hospital settings in other parts of the world. This author has recently received inquiries about providing support for future pediatric medical DMT program in a few countries, including China, Canada and Sweden. In 2016, three members of the MSKCC pediatric integrative medicine team, this author, music therapist Karen Popkin and yoga instructor Catherine Genzler were invited by the Italian Psycho-Oncology Society in Brescia, Italy to present our work at their congress titled, Complementary Therapies in Oncology: Dance, Yoga, Music, Art. Are There Evidences of Efficacy?

The programs that did existed took several forms with many DMTs holding degrees in allied professions such as social work, counseling, psychology, nursing or psychiatry with post graduate training in DMT. Despite the apparent patient interest and satisfaction with these programs, the main reason these programs no longer exist seems to relate to loss of funding, the short-term group special workshop nature of their design, the dance/movement therapist moving, the DMT activities getting absorbed into other general, other arts-based, child life, or recreational [therapeutic] activities offered at the hospital.

### 9.4. Pediatric Medical DMT—Australia

Elizabeth Loughlin, a dance movement therapist in Australia, has worked extensively with the pediatric population in several contexts. Loughlin offers therapeutic playgroups and individual DMT for depressed and anxious mothers and their infants in the hospital outpatient and psychiatric inpatient settings [64]. She also worked in medical DMT within her social work role in a pediatric hospital setting for over 20 years providing DMT programs for children and adolescents with medical conditions [57,65,66]. The initial dance program was supported by the director who “…saw that expressive dance may address a gap in treating the psychosocial problems in specific life-long chromosomal or hormonal heath conditions” [67] (p. 13). A video *Just Go* created in 1990 about her Turner syndrome (TS) dance group was presented at a European international health conference and was written about in a 1993 issue of the American Journal of Dance Therapy. This video provides evidence of how the dance program she offered for young and midlife women from the TS Support group with this health condition provided them with new ways to think about their condition [57,65].

In a recent article for the Dance Therapy Association Australia Loughlin describes DMT in Australia in the following way “… DMT in Australia is outside the current industrial award and often must rely on serendipity, health staff that know and love the arts, and the availability of funding” [67] (p. 13). This article presents Loughlin’s research about how to creatively sustain and communicate the effectiveness of dance arts programs within settings that employ allied health and education professionals. Loughlin found it was more helpful for a dance/movement program, even a short one, to be situated—embedded—in the particular medical department/clinic/ward so one knew of the medical features of the broad diagnosis that the patients shared, and also a working understanding of the nuances in the hospital workplace that affect the day to day experience of young patients with that pediatric health condition and their families [68].

### 9.5. Pediatric Medical DMT—Israel

Judith Mendelsohn also ran a well-respected pediatric medical DMT starting in 1991, known at the time as the Hadassah School at Hadassah Medical Center in Jerusalem, Israel. Her 1999 article about DMT with hospitalized children, for the American Journal of Dance Therapy is a commonly referenced article in the field. Though the program still exists as an educational school for children with serious medical illness, the recreational activities listed do not include DMT.

From 1981–1998 Ety Shachar Siman-Tov worked at a Therapeutic Center in Tirat-Ha-Carmel organized under the Ministry of Education. It was not a medical setting but serviced children with medical issues including neurological, cerebral palsy (CP), and orthopedic conditions and children with learning disabilities, development disabilities, sensory integration issues, and emotional and behavioral difficulties. There were two dance/movement therapists on staff, along with an art therapist, drama therapist and a consulting medical pediatric doctor. Treatment programming was for ages 4–17 years old and included individual, groups and parents. Siman-Tov states, “During these years children with medical issues were referred to me for DMT treatment. They referred them to me because of my medical knowledge as a Registered Occupational Therapist. They received DMT treatment from me” [69].

### 9.6. Pediatric Medical DMT—Italy

From 2006–2013, Marcia Plevin ran a DMT program on the oncology-hematology unit of the pediatric hospital Bambino Gesù (known as the Vatican Hospital), in Rome. She worked with children going bed-to-bed during their hospitalizations. During that period, she also supervised three dance/movement therapy interns. There were dance/movement therapist’s interested in continuing this program, but it was closed due to lack of funding [70].

Lorena Candini, worked with congenital heart disease from 2008–2012 through the Piccoli Grandi Cuori Association in cooperation with the Unit of Pediatric (Developmental) Cardiology and Cardiac Surgery (U.O. Cardiologia e Cardiochirurgia Pediatrica e dell’età evolutiva) at the Azienda Ospedaliero-Universitaria Policlinico Sant’Orsola-Malpighi in Bologna. Hired as a psychologist, her “…daily work was always led by DMT principles and techniques as if I were in DMT session… [using] touch, movements, relaxing techniques, bodily awareness and attunement, body mirroring, somatic countertransference… as a way of better understanding the patient’s feelings…” [69]. She also supervised dance/movement therapy interns. The initial outpatient program design was ten weekly sessions divided into three age groups (7–8; 9–11; 13–15). It became two groups for ages 8–10 and 13–17. These children and adolescents with congenital heart disease, had been hospitalized, and had experienced heart surgery or other invasive procedures. Families were involved at the beginning of the groups to be introduced to the program and collect information about the children. Parents met in groups regularly with Candini and her team to share about the children’s experiences, progresses and difficulties [69]. At the end of the first year of this program a conference was organized to present the successful results, which were also published in an in-hospital publication [71,72].

## 10. Advances in Pediatric Medical DMT within Integrative Oncology—Future Steps

The last section addresses the next steps for pediatric medical DMT highlighting why and how it is an essential therapeutic modality within the integrative oncology team. The focus considers our need to support the psychological health of the growing number of survivors of pediatric cancers; the role of trauma in the body; DMT as an embodied psychotherapeutic modality for trauma prevention; supporting young pediatric patients to regain lost stages of development; educating professionals, patients and families about pediatric medical DMT; and the need for further research.

As this discussion turns to more difficult topics related to potentially painful traumatic reactions and developmental considerations children with medical illness may experience, it is done with utmost respect to the medical professionals deeply committed to healing children without causing harm. The care, kindness and dedication of the medical professionals working with the pediatric population is unprecedented. But many factors contribute to a family and child’s reaction to the medical experience that are out of our control. This section aims to emphasize how the whole medical team, and the integrative oncology team specifically, play an essential role in supporting the quality of life of our patients. The biopsychosocial and patient-centered approach both DMT and integrative oncology effectively addresses, creates an environment to optimize health and empower the patient throughout their cancer care.

### 10.1. Pediatric Cancer Survivor Rates

The National Institute of Health (NIH) National Cancer Institute reports that the survival rate of children with cancer has steadily improved since 1975, when just over 50% of children with cancer under age twenty survived at least five years [73]. Survival rates improved to 83% for this population in a study looking at the years 2007–2013 [74]. The success rate of treatment, though promising, for the child with cancer requires medical professionals to consider how the child’s experience having cancer, and the memories associated with this experience, may affect their later health and well-being.

Counter to the still popular belief commonly stated during medical treatments that the infant or young child is “too young to remember” the medical experience; the long-term psychological influences of the disease must be considered, even if there are no visible disabilities after completing the treatment course [10,23,42]. How much of this attitude may also be related to the difficulty in both recognizing and reconciling that a child, especially a very young one, might be experiencing pain or even trauma, and remembering it? Similarly, the phrase, “he is so resilient”, often discussed with older children and adolescents may belie the complex emotions about the medical experience the pediatric patient may not understand, know how to process, or express. In addition, children and adolescents are actually very good at sensing their parent’s vulnerabilities. Resilient acts can be as much a product of the child’s attempt to support the parent’s feeling of helplessness as the child’s efforts to manage the incomprehensible aspects of the medical experience.

### 10.2. Trauma in the Body-Infancy Memory

Taking the viewpoint that integrating the body and mind creates a body-mind-emotional continuum that begins at or even prior to birth, and informs the infant’s core self from a reality-based perception, leads to the understanding that our bodies hold our experiences [45]. Some experiences are felt and stay in our awareness, but there are also experiences, especially difficult or traumatic ones that are held out of awareness and stored deeply in the body. Gaensbauer, an infancy memory researcher, studying early childhood trauma, documents how sensory experiences and motor actions are linked, and become imprinted. These experiences are organized and represented through “perceptual-cognitive-affective-sensory-motor schemata” [41] (p. 29). These emotional and somatic experiences create a preverbal and sensory-based memory system and can manifest through later clinical symptoms and atypical behaviors including: nightmares; hyperarousal; phobias; anxiety; distress at reminders acting as triggers of the experience; and avoidance of the stimuli related to the traumatic event.

Through his clinical work Gaensbauer demonstrates how [preverbal] trauma is represented in the behavior and traumatic reenactment play schemes of infants, toddlers and children. Gaensbauer states, “…the intimate coupling between sensory experiences and motoric actions that replicate these experiences provides important depth to our understanding of the representations that underlie reenactment behavior” [41] (p. 29). The child’s verbal and nonverbal understanding is separate from their conscious and unconscious expressions of their traumas [75]. Children’s play is more representative of traumatic recall than their ability to verbally discuss their traumatic experiences. Through therapeutic symbolic play, children explore their traumatic reactions creating a coherent autobiographic narrative leading to understanding and healing. Creating a coherent verbal narrative of the traumatic experience enables the young child to make sense of the internalized memory facilitating coping.

### 10.3. Preverbal Trauma and Later Verbal Accessibility

A study about children’s verbal memories related to retinoblastoma treatment specifically before 24 months of age or after 24 months of age and treated before 5 years old, sheds light on the need to use nonverbal methods to access early childhood memories [76]. This age distinction highlights the transition between pre-verbal and verbal language and was used in this study to “understand the relationship between age and verbal accessibility of traumatic memories” [76] (p. 566). Infants’ abilities for verbal recall of events prior to language continues to be a contested debate. The results demonstrate that though both age ranges initiated verbal recall of the medical event, the older children were more able to access their memories of the experience verbally and through the props provided in the free-play task. The researchers conclude that children undergoing retinoblastoma treatment in the first year of life may experience “misconceptions or gaps of knowledge” which may affect their later quality of life, for they “have limited access to a dialogue about these events in their early life” [76] (p. 576).

Susan Coates, a psychoanalyst well known for her work with young children and trauma, discusses whether early traumas experienced by preverbal children are available for symbolic interpretation later in treatment [77]. She concludes that toddlers, infants and neonates experience pain and demonstrate traumatic symptoms. These events can be stored in memory, can be symbolically represented and can affect later behavior and learning. Coates states that resolving the trauma is definitively affected by the quality and function of the attachment system.

### 10.4. Trauma in the Body—Implications Later in Life

Recent research about adult chronic pain, medically unexplained symptoms (MUS) and somatization spectrum disorders (SSD) state that children that have undergone painful or traumatic early medical experiences are more vulnerable to later chronic pain conditions as adults [12,13,78]. Current research and literature in trauma also confirms that trauma is stored in the body causing physiological changes [12,13,15]. Healing methods that enable a reintegration of body, mind, and emotions include therapies that incorporate body and movement-oriented activities.

### 10.5. Pediatric Medical DMT and Trauma Prevention

The intrinsically body and movement-focused aspects of pediatric medical DMT align with these suggested methods. The in-depth nonverbal foundation of DMT coupled with children’s natural propensity for creative expressive dance and movement uniquely position DMT as a treatment method that can support a child to synthesize potentially traumatic aspects of their medical experience while they are actively undergoing treatment. Might this ability to express feelings while in treatment, within the context of an embodied psychotherapeutic milieu create perceptual-cognitive-affective-sensory-motor schemata that foster expressivity and empowerment rather than internalized representations of trauma?

### 10.6. Why Pediatric Medical DMT is an Essential Part of the CANCER-WWW Team?

Pediatric medical DMT and MSDMT enables the patient and family to explore their experience of their illness while it is occurring. The embodied focus, attention to subtle nonverbal cues and approach coming from the perspective of the body-mind-emotion continuum provides an immediate and safe psychological environment that is meaningful and enjoyable. The physical act of playful dancing literally mobilizes feelings and revives the pediatric patient’s natural avenue of expression. Utilizing this innate form of expression during the treatment may prevent the unspoken [traumatic] aspect of the medical experience from being held in the body. This author proposes that pediatric medical DMT can be a form of trauma prevention. This is clinically confirmed, each time a patient returns for their yearly check-up and speaks with warm reverie about their playful dancing experiences delightfully recounting specific details of our dance activities, seemly without awareness of the difficult treatment experience going on simultaneously. A parent shared that one afternoon while the family was in a music store her child pointed to a rainstick—an instrument frequently used during MSDMT to create a soothing auditory sensation—and told her sister it was a toy for relaxation.

### 10.7. Strengthening the Presence of Pediatric Medical DMT—What it Will Take?

Though we can never know fully, the profound biopsychosocial experience of the pediatric patient and family’s cancer journey, as [mental] health care providers we strive to make that passage smooth, and painless, embracing the patient and family with love, care and respect. Through DMT we can address the wordless experience, creating an *embodied coherent narrative* to support the patient and family’s cancer voyage. A specific path must be taken to strengthen this role for dance/movement therapists working with the pediatric population.

First, a strong understanding of infant mental health is necessary. This training enables the dance/movement therapist to recognize and identify how the attachment system can best support the specific presenting family dynamics. Understanding the role of the parents and supporting their experience so they can be effectively available for their child is vital.

It is essential to know the influence the felt-body experience has at all levels of development starting in infancy and continuing throughout childhood and adolescence. This includes awareness of multi-sensory sensitivities that can develop and greatly affect the child’s ability to maintain emotional and physiological regulation. This affects the child’s social/emotional presentation, behavioral state and coping during their treatment, enabling the dance/movement therapist to sensitively attune to the amount of sensation and input that best supports the patient’s engagement. In-depth comprehension of infant and early childhood mental health also provides an understanding of the metaphoric and symbolic levels of nonverbal expression specific to the developmental stages of childhood that also influence adolescence.

The role of early life experiences as it effects behaviors and later life stages is important. Starting with infancy development leads to childhood and adolescent development. A strong understanding of the social/emotional, physical and cognitive themes as they manifest in verbal and nonverbal behaviors is essential for the dance/movement therapist working in pediatric medical illness. Crucial to this training is an understanding of how children and adolescents at different stages of growth comprehend, experience and symbolically represent life and death both verbally and nonverbally. Cultural, religious and ethnic differences related to end of life beliefs must be considered.

Next, is educating the medical team, patients and families about DMT. The psychotherapeutic aspects of DMT must be explained to enable dance/movement therapists to implement the full breadth and depth of their skills, differentiating DMT from other complimentary yet distinct dance and movement-based programming. This will enable more appropriate referrals and effective participation in the Cancer-WWW team.

Increased awareness about the significant contribution pediatric medical DMT has in the integrative oncology team will facilitate dance/movement therapists being hired specifically as dance/movement therapists. This will expose the unique and salient contributions that a dance/movement therapist can make, rather than adding this perspective through the lens of an allied profession such as social work, psychologist, child life specialist, or recreational therapist.

The last imperative is research. With the increasing need for evidence-based practice developing research tools that effectively analyze both the art and science of pediatric medical DMT is essential. Goodill, Koch, Cruz, and others in the field advocate for a research paradigm that synthesizes both quantitative and qualitative methods [18,24,58,79]. Examining its efficacy can wait no longer. Each pediatric medical DMT article discussed in this paper starting in 1993 has stated the need for research. Madden et al.’s research on the creative arts therapies used to improve the quality of life of pediatric brain tumor patients is the only research to date that includes pediatric medical DMT in their study design [23].

1993 was twenty-five years ago. It is time for a call for action. Disallowing treatment protocols due to a lack of evidence-based research is a growing reality. Pediatric medical DMT cannot take an influential place on the integrative oncology team with only clinical narratives as its evidence.

## 11. Conclusions

Pediatric medical DMT is a body-mind-emotion psychotherapeutic approach that supports the patient and the whole family enabling them to express thoughts and feelings about their medical experience using many dance, movement, music, play, and mindfulness techniques. Through this method patients can explore all their emotions along the spectrum from joyful to fearful and rage, creating a sense of empowerment, while attuning to their bodies. DMT also supports body awareness, increases mobility, flexibility, and range of motion, and supports regaining skills in all areas of development. The emotional, felt-body focus supports all stages of the cancer experience from treatment to palliative and end-of-life care.

With the current understanding of the integral relationship between the body-mind-emotions, the psychotherapeutic perspective and the intrinsically nonverbal and embodied nature of pediatric medical DMT, it is uniquely positioned to be a strong member of the integrative oncology team. The extensive definition of terms, literature and research review, and discussion about the national and international state of pediatric medical DMT aims to clarify and define the field, placing it as a central psychotherapeutic treatment modality in integrative oncology that supports the biopsychosocial experience of the pediatric oncology patient and family members. As stated by a parent of a pediatric patient that extensively used pediatric medical DMT and specifically MSDMT methods, along with other integrative medicine services at MSKCC, *“Integrative medicine didn’t change the outcome for us, but it did change the story”* [80].

This paper ends by reiterating a central question: Can pediatric medical DMT support the patient to express feelings while in cancer treatment within the context of a psychotherapeutic milieu, enabling the patient to create an embodied coherent narrative that fosters expressivity and empowerment rather than internalized representations of trauma? If this clinical hypothesis is true, postponing the research to prove this theory is denying services that can greatly contribute to the quality of life for all pediatric cancer patient and family members.

## Figures and Tables

**Figure 1 children-06-00014-f001:**
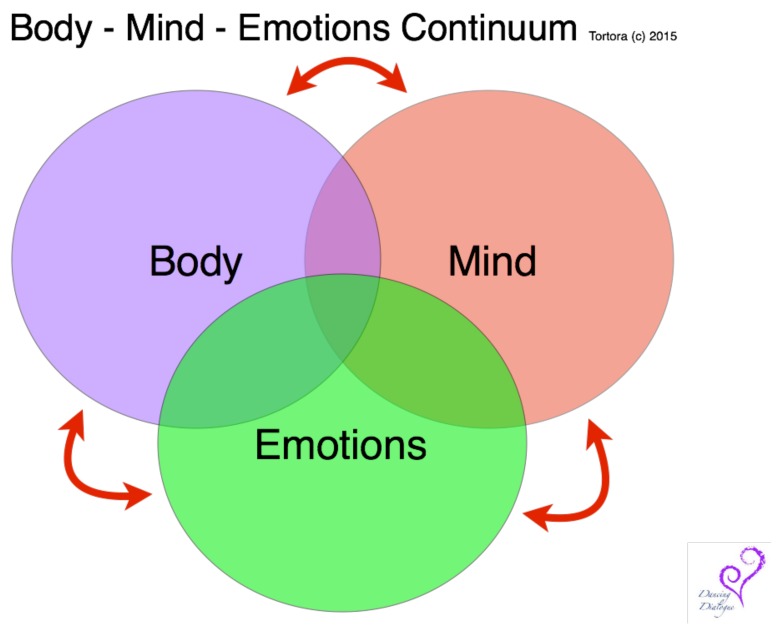
Tortora’s image of the continuous, circular connection between each aspect of the self.

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
