# Peer review of "Children Are Born to Dance! Pediatric Medical Dance/Movement Therapy: The View from Integrative Pediatric Oncology"

_children, 2019, doi:10.3390/children6010014_

Round 1
Reviewer 1 Report
Based on the author’s extensive experience working with colleagues at the Memorial Sloan Kettering Cancer Center, the paper describes how Dance Movement Therapy (DMT) can be integrated into the overall treatment of children diagnosed with various forms of cancer. It is of particular interest because it focuses on aspects of children’s experience with serious medical illnesses, such as cancer, which are not ordinarily addressed in traditional treatment protocols.
The paper is well written and covers a wide range of relevant issues. It provides historical and scientific background outlining the value of non-verbal, emotionally centered, movement based approaches in the treatment of medically ill children. In particular, developmental research documenting how embodied, non-verbal memories are at the core both of children’s sense of self and their relationships with others is reviewed, as well as the ways in which an understanding of these processes of embodiment can be utilized in the service of understanding and treating patients’ experience with cancer and its treatment. In conjunction with this background review, the importance of the child-parent relationship for adaptive coping in the face of adversity is also underlined. Clinical examples are provided that illustrate how such approaches to child patients at different ages and their families can elicit feelings and thoughts that might otherwise not be expressed while at the same time establish therapeutic vehicles for helping patients and families to cope with the stresses associated with a severe, painful, life threatening medical condition. The distinguishing elements of dance movement therapy (as compared to other forms of dance) are outlined and the current status of dance movement therapy is also discussed, including where in the world it is practiced, the challenges associated with establishing DMT programs within medical settings, and the importance of carrying out systematic research regarding its effectiveness.
While the paper can certainly stand as it is written, I would have a couple of suggestions for strengthening the paper. The detailed clinical vignettes were extremely effective in helping the reader to picture how the therapeutic principles outlined in the text are actually applied in the clinical setting. While the interventions were clearly helpful in and of themselves, what was not described was how the particular insights and therapeutic interventions gained through individual DMT work were communicated and integrated into the patient’s overall treatment. Both in regard to the individual cases and more generally, more detailed description of the ways in which DMT therapists are likely to interact with other members of the treatment team and how clinical information provided by the DMT therapist is likely to be utilized in the overall treatment program would be helpful to the reader by placing DMT work in its larger medical context. I also thought that, given the usefulness of clinical examples in bringing the points the author was making to life, that any further vignettes the author might like to add would be very welcome.
Finally, there were a few places in the text where the same points were made on more than one occasion or where different aspects of the same issue were discussed at different points in the paper when they could have been put together (an example of the latter would be where discussion of the differences between DMT and traditional dance activity occurs at two different places). Some minor tightening and/or rearranging of the text might enhance its overall flow and internal coherence.
Author Response
See uploaded file

Reviewer 2 Report
This is an excellent and much needed review paper on the application of dance/movement therapy to a pediatric oncology population. The paper is well written and generally well organized. The author is a world renowned expert in this field. I think to strengthen this already important potential contribution to the literature on pediatric medical trauma and psycho-oncology, the following points might be considered by the author.
An important theme in the literature that only emerges on p. 4, l 158-161, to this reviewer's mind should be mentioned in the very beginning of the paper. Namely, the notion that dance and creative movement involves a generally pleasurable control of one's body which is particularly therapeutic for chronic illness patients, among whom even more so for oncology patients, who have the sense that their body and their disease are "out of control." Since nearly all treatments for traumatic stress involve mastery and control over events and sequelae that are out of the individual's control, this literature on traumatic stress intervention could also already be cited along with references about the idea that offering solutions to feeling helpless and reimagining one's body's potential to strive for health and comfort is essential to trauma-focused treatments. See Robert Pynoos' work and others. A place in the text that could be appropriate to introduce this central theme would be on p. 1, l 33 after "...ability to participate actively in playful childhood activities is frequently compromised." (best not to split infinitive in that sentence) There one could introduce something to the effect of: "This loss is akin to the sense of disempowerment and loss of control of one's body and stress physiology when facing a threatening, traumatic life event.."
The clinical examples are compelling and illustrative of the points raised in the review. However, for non-MD readers, this reviewer is concerned about the use of the diagnosis terms such as "plexus carcinoma" or "retinoblastoma" or "leukemia" (so vignette sections 8.1, 8.2. and 8.4). Each of these diagnoses to an MD will present a host of associations. The author might put a footnote to explain these diagnoses and in the case of leukemia, specify which kind if pertinent.
The author might also find a place to include how the intervention impacts the child-caregiver relationship and indeed, when the caregiver(s) and/or staff might be involved in the intervention at different ages, and how (i.e. as participants, as audience) to maintain, maximize and augment the therapeutic effects. This and the application of dance/movement to form a non-verbal form of "trauma narrative" would also be of interest.
Author Response
see uploaded word file

Reviewer 3 Report
This article addresses an important topic, of the function and value of DMT in paediatric care. It offers much theoretical material, compelling case studies, a literature review and some research the author has undertaken to substantiate her argument. The author is evidently a highly skilled and knowledgeable professional in this field who draws on extensive experience to illustrate this writing.
However, the article in its current state is not suitable for publication. It is far too long. Given one intended audience of busy medical professionals, this is counter- productive to their likely engagement with the material. The written expression is rough and gets rougher as the article goes on. Much of it does not conform to academic conventions, especially in latter sections. Detail needs to be reduced, especially where it does not offer any enlightenment. References section is rough.
I include below many suggestions for improvement, but only until line 557. The next part needs even more work.
Restructuring some sections is also necessary. The literature review that appears after case studies should be in lit review section. Likewise, the method for gathering information discussed after the case studies should be in method section. The librarian need not be mentioned here but only the lit review undertaken. The acknowledgement of their support should be in appropriate section.
0: natural
19: a biopsychosocial psychotherapeutic perspective- this sounds like everything and therefore perhaps nothing…
33: maybe a different word best in this sentence: activity and actively
35: I think it might be ‘dance movement psychotherapy’
38: Why did you choose these two outcomes of DMT, amongst the many?
‘to enable the patient to express feelings 39 and gain a deeper understanding of experiences that may be difficult to speak about’? are they particularly relevant for this use of DMT?
44: DMT with children with medical illness is a little known yet growing application in the field: I don’t think children with medial illness is an application, but a population group and context. Maybe rephrase somehow.
89: The field evolved from modern dancers in the 90 forties and fifties that recognized the emotional healing powers of expressive dance [2].
As it is recognised in the USA, the field evolved from the work of modern dancers in the 1940s and 50s who recognized the emotional healing powers of expressive dance
95: no comma, something missing here; what has Goodill added here, extra aspect to the definition?
95: treatment goals for this group? Improved management of pain is a goal, techniques are not
115-122- refs
141: don’t use foundation twice in sentence
155: through use of a
157: full stop before starting new sentence: this includes
160: soothe
174: what do you mean by ‘each definition?’
187: philosophy with ideas from the work of classic etc…
214: at that moment?
225: this is the first time spiritual is used. It seems a bit incongruous to start here without mentioning this new edition to the paradigm.
238: maybe ‘DM therapists’ rather than ‘we’ here
244: I’m not sure about this claim- might be better to demonstrate why DMT is essential rather than just claiming it.
260: substantiates the premise that …
265: comma after vocabulary
269: maybe in this section also discuss what happens in a dance session or other movement modality that is different
290: through the ADTA or accredited/recognised by?
288/289: Masters, masters, master’s ?
293: should be mental health counseling, marriage and family counseling and social work.
297: comma after MCKCC
303: esthetic? Aesthetic?; no comma after relaxation
308: the child’s development
316: well cared for, (comma)
317: process their experiences; competence?
321: is the author’s belief relevant? Maybe better, understanding
324: no comma after define
335: consistency with referencing. Eg. [34] (pp. 65). and [1, 6, 36-39].
345: Re 7.2. Social, Religious, Cultural and Ethnically Responsible Practice
Is ‘responsible’ an accepted term here?:
349: ‘the’ specialization rather than ‘our’, more professional…
350:no comma after neurob.
358: need an intro sentence to say what LMA is, why you are talking about it here and reference- and move it closer to 7.1. Use capitals for LMA elements
qualitative
359: comma before Effort
369: why does it matter that they are in Italy?
368: introduce the concept of counter transference and why we would need to hear about it before the example. Move up text from 376
372: essential tools
373: attention to etc…. is a core process…
384: no comma needed, that provides?
389: delete comma
390: not really a session, but series
391: beautifully and carefully described program
448: comma after half
448/452: Mom or mom
453: move comma
470: comma after hurdle
473: inverted commas don’t have a beginning
480: do you mean you have recorded lion roars that you play?
496: what does it matter that it is Monday morning?
500: complementary
501: participants’
502: music making,
524: comma after warrior
527: caps in child life
525: this is not just a session either
558: what are ADTA and EADMT: maybe DMT professional associations in Europe and USA
560: third person here but you have used first person elsewhere
575: research in all DMT populations is lacking due to the nature of a DMT session. I’m not sure this is the reason research is lacking
580: undertake research
557: this section is quite under-developed. Suggest more formal academic approach. Who did the search is not relevant there, but only what they found. Then this person can be acknowledged appropriately elsewhere.
This lit review should come earlier in lit review section’
Author Response
see attached word file

Round 2
Reviewer 3 Report
The article is shaping up well. It offers a comprehensive overview of an important topic by an author who is clearly committed and informed. It is much improved since my last reading in form and structure.
I am pleased to see that the author has responded to most of the corrections I have suggested. Unfortunately not all, including the opening line which still doesn’t make sense. And other new mistakes are there. It would have been far preferable for me not to spend so much time reviewing because of so many basic grammatical and English
The article is shaping up well. It offers a comprehensive overview of an important topic by an author who is clearly committed and informed. It is much improved since my last reading in form and structure.
I am pleased to see that the author has responded to most of the corrections I have suggested. Unfortunately not all, including the opening line which still doesn’t make sense. And other new mistakes are there. It would have been far preferable for me not to spend so much time reviewing because of so many basic grammatical and English expression mistakes that prevented me from reading easily. The software Grammarly could be helpful for future writing.
I haven’t suggested that the author should be writing a research article, but if she is writing an article for an academic professional audience, then it must be a minimum standard professional quality of writing. It is much improved now, especially the last section.
I think the article is still too long and includes content that could be reduced without loss of information, but presumably it meets the journal’s word limits. Shorter articles are more accessible for the busy medical professionals the author would presumably want to influence by reading the article.
10. Children freely expressing themselves through spontaneous dance is a nature part of childhood. Should be natural?
37: needs space between dancemovement
71: whichcan
94: origins
99: this direct quote doesnt have a page number next to it, but previous like quotes do. Make this consistent throughout the article
103: that should be who
108: no comma after Goodill
113: extra semi colon
116: acceptance
119: needs comma at the end
144: this claim ‘building strong relationships throughout the full course of the treatment’ should be ‘which can contribute to stronger’
153: fullstop in wrong spot
175 needs comma after analgesic
176: needs ‘and’ after sensation
186 and 189 are repetitive
208: nice point!
224: needs comma after illness
262 no comma after yoga
305: I’d suggest this statement ‘These are extremely useful skills’, might be tempered as’ these can be useful’
312: witnessing isn’t defined or explicated here yet it is fundamental to the point of the paragraph.
329: not counsellors but counseling and social work
335: Im not sure that the profession of DMT is best advanced by considering availability of the therapists as a ‘goal’
348: this ref looks wrong [1, -] (pp. 31).
356: two fullstops
377: Explain what LMA and KMP stand for
378: consistency with capitals on LMA terms. Add at least one ref for LMA. Need to introduce what LMA is: a movement analysis system
380: should be qualitative
389: why italics?
393: no – needed
400: is DMT the modality or the therapist?: I suggest use DM therapist
401: add ref for counter transference when you first mention it
Comma not needed in this sentence
417: no dash
442: do you perform medical procedures at other times?
444: with respect to?
446: maintaining
450: comma after ‘session’ but not ‘though’
456: capital on Mom
460: comma after quiet
Appears to have had a calming effect?
476: Mom
490: commas: over, Mom looks up with a pleading
513: no comma: I know he knows
518: singular or plural: recordings, it?
534: why no italics here. Music therapists Karen Popkin
The article is shaping up well. It offers a comprehensive overview of an important topic by an author who is clearly committed and informed. It is much improved since my last reading in form and structure.
I am pleased to see that the author has responded to most of the corrections I have suggested. Unfortunately not all, including the opening line which still doesn’t make sense. And other new mistakes are there. It would have been far preferable for me not to spend so much time reviewing because of so many basic grammatical and English expression mistakes that prevented me from reading easily. The software Grammarly could be helpful for future writing.
I haven’t suggested that the author should be writing a research article, but if she is writing an article for an academic professional audience, then it must be a minimum standard professional quality of writing. It is much improved now, especially the last section.
I think the article is still too long and includes content that could be reduced without loss of information, but presumably it meets the journal’s word limits. Shorter articles are more accessible for the busy medical professionals the author would presumably want to influence by reading the article.
10. Children freely expressing themselves through spontaneous dance is a nature part of childhood. Should be natural?
37: needs space between dancemovement
71: whichcan
94: origins
99: this direct quote doesnt have a page number next to it, but previous like quotes do. Make this consistent throughout the article
103: that should be who
108: no comma after Goodill
113: extra semi colon
116: acceptance
119: needs comma at the end
144: this claim ‘building strong relationships throughout the full course of the treatment’ should be ‘which can contribute to stronger’
153: fullstop in wrong spot
175 needs comma after analgesic
176: needs ‘and’ after sensation
186 and 189 are repetitive
208: nice point!
224: needs comma after illness
262 no comma after yoga
305: I’d suggest this statement ‘These are extremely useful skills’, might be tempered as’ these can be useful’
312: witnessing isn’t defined or explicated here yet it is fundamental to the point of the paragraph.
329: not counsellors but counseling and social work
335: Im not sure that the profession of DMT is best advanced by considering availability of the therapists as a ‘goal’
348: this ref looks wrong [1, -] (pp. 31).
356: two fullstops
377: Explain what LMA and KMP stand for
378: consistency with capitals on LMA terms. Add at least one ref for LMA. Need to introduce what LMA is: a movement analysis system
380: should be qualitative
389: why italics?
393: no – needed
400: is DMT the modality or the therapist?: I suggest use DM therapist
401: add ref for counter transference when you first mention it
Comma not needed in this sentence
417: no dash
442: do you perform medical procedures at other times?
444: with respect to?
446: maintaining
450: comma after ‘session’ but not ‘though’
456: capital on Mom
460: comma after quiet
Appears to have had a calming effect?
476: Mom
490: commas: over, Mom looks up with a pleading
513: no comma: I know he knows
518: singular or plural: recordings, it?
534: why no italics here. Music therapists Karen Popkin
539: participants’
546: Pierre, age 7,
567: once he is out of bed
596: intricate? Fundamental? Intrinsic?
605: patient,
616: hesitant
621: devices
631: do you mean in DMT?
645: soothe, caregivers
670: capitals on Child Life?
681: is used
684: toddler who
686: ease
705: comma after only
714: use of tools from verbal interventions that
716: I’d suggest, ‘very few researchers in DMT’
721: doctoral
739: match verbs here, decreased, helped/helping
752: I think this is too much info re this family.
785: delete ‘this past’, as it won’t be in future when people are reading the article
805: DM therapists
826: Dance Movement Therapy Association of Australasia,
855: children,
856: DM therapists, no apostrophe
I don’t know this referencing style so I can’t check the referencing- but presume that will happen in the editing process.
858: no comma after Candini
867: no comma after disease
875: steps that I recommend; DMT,
978: maybe ‘recounts with delight’? Does this happen with every client? Can you indicate here how often this occurs? It seems a very big claim
983: no comma after fully
1002: something missing in this sentence: Starting with infancy development leads to childhood and adolescent development.
1011: should be complementary
1019: comma after practice
1041: grammar issue: DMT can’t be a ‘member’ of a team
Are you talking about this article here? Clearer if you said so.
Re conflict of interest: I think it might be prudent to name here that the author has a long term career investment in this form of DMT that she is discussing.
While this investment is also clear throughout the article, and is a strength, this position does make it impossible for her to offer a dispassionate assessment of the contribution of her field, and this lack of dispassion is also evident in the article, as per some of the suggestions I have made for reduction of claims about efficacy of the work..
I'd suggest considering whether Sherry Goodill would want herself named as a leading researcher. She is a leading advocate for DMT research and educator and has publishedmuch relevant material work over her career, but right now and for the foreseeable future she is not doing research. This claim might be counter productive for the DMT field and Sherry.